# Gender differences in job satisfaction among disabled workers

**Seunghee Yu[1], Chung Choe[2]***

**1** Department of Social Welfare, Sungkyul University, Seoul, South Korea, **2** Department of Economics, Konkuk University, Seoul, South Korea

\* choechung@gmail.com

## Abstract

This paper explores the association between job satisfaction and gender for workers with disabilities, using data from the Panel Survey of Employment for the Disabled, which interviews officially registered persons with disabilities in Korea. To take full advantage of the longitudinal data, we apply random-effects ordered probit models to investigate the underlying factors that affect gender differentials in job satisfaction. Our findings reveal that merely different work values between women and men do not account for the significantly higher job satisfaction among women. We suggest that workers' expectations play a role in explaining why female workers are happier in the workplace than their male counterparts; that is, holding other factors constant, women's expectations from jobs are lower than men's. This hypothesis is *partially* supported by the empirical analyses that gender differentials diminish among the highly educated workers, for whom there is less likely to be a gender gap in terms of job expectations.

## 1. Introduction

Over the last several decades, job satisfaction has been of interest to social scientists because it is closely related to conditions in the work environment and important in explaining worker productivity, absenteeism, and resignations. Numerous studies have attempted to investigate various determinants of job satisfaction, including gender [1, 2], age [3], pay [4], characteristics of the work environment [5], and job matching [6]. Among other factors, the relationship between job satisfaction and gender is noteworthy, since job satisfaction is greater for women than for men despite a longstanding gender gap in the labor market that favors men [1, 2, 7, 8].

Many studies have tried to explain the implications of the gender gap in job satisfaction using personal and job-related characteristics and subjective factors such as men's and women's different work values, relative utility, and expectations [2, 8–10]. According to their findings, objective variables, such as personal and job-related characteristics, help us to understand what type of worker is satisfied with what type of job, but not why women are more satisfied than men. Rather, subjective variables, such as men's and women's different attitudes, values, and evaluations of jobs, were presented to explain the gender differences in job satisfaction [1,

**Data Availability Statement:** The data is available at: https://edi.kead.or.kr/ENG_Index.do. The official dataset name is "Panel Survey of Employment for the Disabled".

**Funding:** The author(s) received no specific funding for this work.

**Competing interests:** The authors have declared that no competing interests exist.

2, 8]. Clark explained gender differences in job satisfaction in terms of relative utility [2]. He argued that women have lower expectations because their jobs have been so much worse in the past than men's, and those who expect less from jobs will be more satisfied with them.

The earlier work on job satisfaction has almost exclusively analyzed nondisabled workers. However, it is worth exploring the topic of job satisfaction with a focus on workers with disabilities because of their unique circumstances in the labor market: low wages, high unemployment rate, segregation in low-paying occupations, and productivity limitations. While these intrinsic differences in circumstances are likely to generate a different level of subjective well-being among workers with disabilities, only a limited number of studies have focused on their job satisfaction, and the focus has been on comparisons of job satisfaction between nondisabled and disabled workers [11, 12]. There is less evidence on how disabled workers feel about their job and workplace. Therefore, the present paper aims to explore variables in determining job satisfaction among disabled people, giving particular attention to gender.

One of the primary restrictions on research on disabled people is the unavailability of datasets. In particular, most datasets on labor-market outcomes have an insufficient number of observations for persons with disabilities. In this paper, we use a novel dataset, the Panel Survey of Employment for the Disabled (PSED), to explore job satisfaction among workers with disabilities in South Korea. In contrast to other typical labor-market surveys, where information is collected for both disabled and nondisabled individuals, the PSED includes only persons with a certificate of disability registration. This unique dataset allows us to access a large sample size of workers with disabilities. Furthermore, in the PSED, the disabled population is defined in a more objective way than in other datasets, in which disability is defined by self-reported health conditions. Taking advantage of the PSED, we aim to analyze the effects of gender on job satisfaction among workers with disabilities, giving particular attention to subjective variables, such as work values and expectations, as well as objective ones (personal and job-related variables).

We contribute to the literature by providing representative empirical evidence on the determinants of job satisfaction for an important sub-group of the population that has been largely neglected in the literature. Our results indicate that women with disabilities show higher job satisfaction at work than men with disabilities do, a finding similar to that for nondisabled workers. Section 2 reviews prior studies on job satisfaction. Section 3 describes the data and statistical model used in the analyses. Section 4 presents our results, and Section 5 concludes, addressing some limitations.

## 2. Literature review

### 2.1. Job satisfaction and gender

Employees' job attitudes provide useful information about their inclusion and productivity in an organization [13, 14]. Employee job satisfaction refers to "a pleasurable or positive emotional state resulting from the appraisal of one's job or job experiences" [15]. Thus, job satisfaction can be defined as the utility that people gain from their work [16]. Recent empirical analysis has considered employees' own perceptions of their job, since these are related not only to social inclusion but also to employee performance, absenteeism, organizational commitment, and turnover [17–20]. Thus, an understanding of employees' subjective well-being can provide important information for understanding their labor-market behaviors.

Results of empirical research on the differences in job satisfaction between male and female employees have reported that women exhibit higher job satisfaction than men do, despite their less favorable work outcomes (e.g., lower wages, labor-market discrimination, and labor-market segregation) [2, 9, 10, 21]. Numerous studies have explored the source of this gender

paradox and have raised several possibilities to account for it, such as differences in personal and job characteristics, selectivity bias, work values, relative utility, and expectations [1, 2, 7].

Clark has confirmed that many individual and job-related characteristics differ markedly by gender, but gender differentials in job satisfaction are still significant even after controlling for those variables [2]. One argument related to selectivity bias suggests that dissatisfied female workers leave the workplace more easily than equally dissatisfied male workers and thus that the remaining female workers are likely to have higher job satisfaction. However, Clark's and Sanz de Galdeano's findings indicate that selectivity bias does not explain the gender–job satisfaction gap [2, 22].

The personal characteristics associated with job satisfaction are not only objective (i.e., age, education, income, etc.) but also subjective, such as an individual's priorities or values. Clark found that men's and women's work values are significantly different: men are more concerned about the extrinsic aspects of work, such as pay and promotion, whereas women are more likely to consider the intrinsic aspects of work, such as relations with coworkers, the actual work itself, and the hours of work [2]. It appears that work value variables are important predictors of job satisfaction: workers who had said that promotion or pay was important reported significantly lower job satisfaction, while those who placed a higher priority on relations at work reported higher job satisfaction. The gender gap in job satisfaction decreased when controlling for these variables.

Researchers have also identified the presence of relative terms in the utility function; that is, men and women are faced with different reference groups or relative incomes [1, 2, 4, 23]. In other words, women are more satisfied with their jobs because they expect less than men do or compare themselves to different reference groups when evaluating their jobs. The concept of job expectations refers to what workers feel they should receive from work, which may be partially determined by relative, rather than absolute, variables [1, 2].

According to Crosby, most workers compare themselves to their counterparts of the same sex when evaluating their jobs [23]. The higher job satisfaction among women would reflect their lower expectations, which would result from the poorer positions in the labor market held by comparable women. Clark further investigated other factors that are likely to be correlated with job expectations [2]. It turned out that the gender gap in job satisfaction was not observed among young, highly educated workers and those in professional occupations, likely because such female workers are less likely to have lower expectations even when compared to their male counterparts. These earlier studies confirm that the gender gap in job satisfaction can best be understood by a combination of subjective variables (work values and expectations) and objective variables (personal and job-related characteristics). In a more recent study, Perugini and Vladisavljevic explored the gender–job satisfaction gaps in 32 European countries, showing that women employed in typically male occupations are more likely to have expectations akin to those of their male counterparts, while higher educational attainments do not have a similar effect [24].

## 2.2. Job satisfaction and disability

It has been reported that workers with disabilities experience poorer labor-market outcomes in terms of employment, earnings, and job quality [25, 26]. Disabled workers tend to be overrepresented in bad jobs, whereas nondisabled workers are more likely to be overrepresented in decent jobs [16, 27, 28]. There is also growing evidence of the gap in employment and earnings between disabled and nondisabled workers [29–31]. It turns out that the average employment probabilities of disabled people in Britain were estimated to be 40 percentage points lower than those for their nondisabled counterparts, holding other factors constant, such as their

demographic and economic characteristics [30]. Although earnings gaps are narrower in comparison with the gap in employment rates, earlier studies report labor-market discrimination against workers with disabilities, leading to wage gaps depending on disability status [26, 31]. Employers may not hire disabled people if they underestimate their productivity or exploit their geographical or occupational immobility by paying lower wages [32–34].

Exploring the effect of disability on labor-market outcomes has involved objective features, such as employment rate [30], type of employment [35], hours of work [36], and wages [26, 31]. On the other hand, we observe less research on the subjective well-being of workers with disabilities, such as job satisfaction. However, understanding the difference in job satisfaction for disabled employees is crucial for both employers and policy makers who hire and support workers with disabilities [16, 20]. It should help them by providing insights into the effectiveness of employment policies and practices for disabled workers.

The results of empirical research on the job satisfaction of disabled workers support two contradictory conclusions [12, 16, 37, 38]. The first view on the relationship between disabilities and job satisfaction is that disabled workers demonstrate higher levels of job satisfaction than do nondisabled ones, ceteris paribus. Pagán and Malo argued disabled workers' higher job satisfaction might reflect their lower expectations, which likely result from their disadvantaged position in the labor market [12]. Thus, an actual-aspiration gap model postulates that the closer people's actual experienced situations are to their expectations, the higher their level of satisfaction will be [2, 39].

The second perspective is that people with disabilities are less satisfied with their jobs than are nondisabled counterparts. Perales and Tomaszewski demonstrated that most disadvantaged groups—i.e., those who are female, very old or very young, non-white, homosexual, or non-degree-educated—are likely to be more satisfied than advantaged groups are, given a similar job [16]. Disabled workers, however, rated the same proximate-job conditions more unfavorably than nondisabled workers did. Baumgärtner et al. found that employees with disabilities were less satisfied than were their colleagues without disabilities in highly centralized working environments, whereas they were more satisfied with their jobs in decentralized workplaces, implying that structural flexibility is important for workers with disabilities [38]. Cox and Blake [40] and Schur et al. [41] reported that disabled employees are less satisfied because they face many work-related challenges: 1) disadvantages in terms of pay, training, and decision making [41], 2) disability-related restrictions as a result of non-optimal or non-granted accommodations [42], and 3) working experiences that are less positive as a consequence of discrimination [43]. The disadvantaged positions of disabled employees in the labor market may be reflected in a more negative view of their company, job satisfaction, and company loyalty [20].

Disabilities and gender (female) are socio-demographic traits that have historically been associated with disadvantageousness and subordination in the labor market. One recent study, using Spanish panel data, has addressed the determinants of job satisfaction [12] with emphasis on gender differences in satisfaction between workers with disabilities and their nondisabled counterparts. The gender gap in job satisfaction was observed among nondisabled workers, but not among workers with disabilities. Female workers with disabilities did not report statistically significant higher levels of job satisfaction than did males with disabilities. In line with Clark's interpretation [2], the gender gap in job satisfaction for nondisabled workers was attributed to women's lower expectations. On the other hand, the authors suggested that female workers with disabilities might have similar job expectations to male disabled workers based on the insignificant differences in the disabled workers' sample.

However, there remains the question of why expectations may be the same for disabled males and females. Women with disabilities are in one of the poorest positions in the labor

market, due to being doubly disadvantaged, i.e., in terms of gender and disability [44]. Their poor labor-market outcomes might lead them to have even lower expectations than nondisabled workers or men with disabilities. Therefore, following Clark's argument of expectations in job satisfaction [2], it should be supposed that disabled women have higher job satisfaction than other groups, ceteris paribus.

## 3. Data and methodology

The Human Research and Ethics Committee of Khon Kaen University reviewed and approved this project (registration: HE621484). Written informed consent was obtained from each of the patients in these studies. As for patient records used in this study, all of the data were fully anonymized before their use.

We used the PSED, which is an annual survey that has been collected by the Employment Development Institute of South Korea since 2008. The survey has conducted interviews of 5,092 individuals, aged between 15 and 75 years, who are officially approved and registered as disabled by the Korean government. The data include demographic, socio-economic, disability, and work-related information, with follow-up interviews aiming to track changes in the economic activity of persons with disabilities. In contrast to the typical surveys used for disability studies, which include both disabled and nondisabled individuals, the PSED includes only those individuals with registered disabilities.

Our study period covers seven years, from 2009 to 2015. As typical in longitudinal data, there is data attrition over time in the PSED. The total number of observations has decreased from 5,092 in 2008 to 3,983 in 2015. The 2008 sample is not used in our analyses because occupation information is unavailable for the year.

The labor-market participation rate of people with disabilities is considerably lower than that of others. In particular, since this study is limited to wage workers, about 75% of the observations were dropped out of the analysis sample. We further restricted our sample to those aged between 18 and 65, and after excluding observations with missing information for any relevant variables, the observations in the final sample per year were as follows: 885 in 2009, 974 in 2010, 925 in 2011, 914 in 2012, 846 in 2013, 783 in 2014, and 744 in 2015. The pooled data from 2009 to 2015 contains 6,071 observations: 4,543 for men and 1,528 for women.

The main objective of the paper is to estimate the function of job satisfaction with a particular focus on gender comparisons. In our study, job satisfaction—the outcome variable—refers to an individual's overall emotional state resulting from their appraisal of their job. Therefore, we used the self-reported job satisfaction variable, which is to be measured with a single composite question about job satisfaction. As a measure of overall job satisfaction, workers were asked, "All things considered, how satisfied are you with your present job?" The extent to which workers are satisfied with their jobs was measured on a scale of one to five, where one indicates "strongly dissatisfied" (the lowest level of satisfaction) and five indicates "strongly satisfied" (the highest level of satisfaction).

Given the characteristics of the dependent variable and overall job satisfaction and that the PSED comprises longitudinal data, we adopted a random-effects ordered probit model, which is suitable for dealing with individual unobserved heterogeneity. We expressed the determinant function of job satisfaction in terms of a latent linear response, where observed ordinal responses ($JS_{it}$) are generated from the latent continuous response, such that:

$$JS_{it}^* = x_{it}\beta + v_i + \epsilon_{it} \tag{1}$$

and

$$JS_{it} = \begin{cases} 1 \text{ if } JS_{it}^* \leq \kappa_1 \\ 2 \text{ if } \kappa_1 < JS_{it}^* \leq \kappa_2 \\ \quad \vdots \\ 5 \text{ if } \kappa_4 < JS_{it}^* \end{cases} \qquad (2)$$

where $JS_{it}^*$ is the latent variable of job satisfaction, and $\epsilon_{it}$ is distributed as standard normal with $N(0,1)$ and independent of $v_i$. In the application of the random-effects ordered probit model, the probability that $JS_{it}$ (i.e., the reported job satisfaction) is equivalent to the value $j$ is given by probability:

$$\begin{aligned} \Pr(JS_{it} = j) &= \Pr(\kappa_{j-1} < x_{it}\beta + v_i + \epsilon_{it} \leq \kappa_j) \\ &= \Phi(\kappa_j - x_{it}\beta - v_i) - \Phi(\kappa_{j-1} - x_{it}\beta - v_i) \end{aligned} \qquad (3)$$

where $v_i$ is independently and identically distributed $N(0, \sigma_v^2)$; $\kappa$ is a set of cutpoints $\kappa_0, \kappa_1, \ldots,$ $\kappa_5$ (where $\kappa_0 = -\infty$ and $\kappa_5 = +\infty$); $j$ is the observed $JS_{it}$ with integers from 1 to 5; and $\Phi()$ is the standard normal cumulative distribution function. Using the command, xtoprobit, in STATA 13, we estimated the random-effects model by maximum likelihood estimation.

Empirical analyses are implemented primarily to show the gender differences in job satisfaction and to investigate the factors that might cause such differences. In previous studies, a regression-based approach has been taken to examine the sources of job satisfaction in which one would seek to explore the factor that causes gender differentials if the gap in job satisfaction disappears or decreases once included in the analysis. This study followed the same approach and theoretically speculated on the factors affecting gender differences in job satisfaction. We first investigated gender differences in job satisfaction by including an indicator variable for gender (Female = 1) into the regression models, along with other explanatory variables, such as demographic and disability-related information and job attributes. If our results would be consistent with other studies for nondisabled workers, we would anticipate that female workers with disabilities would report significantly higher levels of job satisfaction than their male counterparts do. Second, we investigated the role of subjective variables as determinants of job satisfaction. We explored the impact of work values on job satisfaction by observing the changes in the gender gap of job satisfaction when including the variables in the regression analyses. The work values in our data were measured by asking the survey participants about the most important aspects of job attributes when choosing a job: pay, job security, working hours, aptitude, work flexibility, etc.

As discussed in the previous section, no earlier work has provided a specific explanation for why female employees are happier at work. Using several different hypotheses, researchers have attempted to explain the paradox that women show higher job satisfaction than men, even in worse working conditions. One such hypothesis concerns relative utility and expectation. In determining job satisfaction, while the absolute level of wages is important, the level of one's wages relative to that of others can also be an important factor in determining the utility of a job. Comparison involves observing other workers and comparing them with oneself, and it is proposed that, as with relative income, relative expectations can play a role in explaining why women's job satisfaction is higher than that of men.

In contrast to the analysis of gender roles above, there is no single explanatory variable to test this hypothesis. Thus, researchers have adopted an indirect approach to examine the role of expectation in the determination of job satisfaction and attributed the gender difference in

job satisfaction to differences in expectations. Clark explains that, for a variety of reasons, the expectations of older, less-educated, and non-professional women will be lower, likely leading to higher job satisfaction conditional on a set of individual and job attributes [2]. In a similar spirit with Clark's study, we conducted regression analyses that include the interaction terms between gender and 1) age (18–35 vs. 36–65); 2) education (high school or less vs. college or more); and 3) occupation (professional/managerial vs. all other occupations). Assuming that, among the young, the highly educated, or workers with professional or managerial occupations, one finds little or no gender difference in expectations, we anticipated that, among these sub-groups, gender does not play a significant role in determining job satisfaction.

# 4. Results

## 4.1. Descriptive statistics

Table 1 provides summary statistics for key variables in the models. The mean of job satisfaction indicates that women are happier at work than men (3.17 vs. 3.14), although the difference is not statistically significant. Male workers are more likely to be educated and to be married or have a partner than female workers. Interestingly, we observe a gender difference in the disability types represented in our sample: male workers are more likely to have physical external disabilities, while female workers are more likely to have sensory disabilities. Female workers report relatively earlier onset of disability than their male counterparts do (21.93 vs 24.95).

With regard to job characteristics, male workers report considerably better labor-market outcomes in many different respects. They are more likely to hold permanent contracts, to work at large establishments, to work for more hours and earn a higher monthly income, and to work in professional or highly skilled jobs. However, we do not observe a noticeable difference in the industry distribution, except that while male workers are dominant in the construction sector, female workers are more likely to work in public/education/health sectors.

To further explore the gender gap in job satisfaction, we also consider the different values that men and women might expect from the workplace. It appears that work values are significantly different by gender, as shown in Table 2. Men tend to place a higher value on pay, job status, job security, and chances for success in the job than women do, while work intensity, work hours, aptitude, commuting distance, and work flexibility are more important factors for women. These findings are consistent with the view that men are more likely to value the extrinsic aspects of work, such as pay and job status, while women are more concerned with the intrinsic aspects of work, such as the working hours and their aptitude for the job [2, 45, 46].

## 4.2. Gender and jobs

Table 3 shows the coefficient estimates of job satisfaction using a pooled sample. While there are no gender differences in unconditional job satisfaction (Model 1), when controlling for demographic characteristics and disability-related covariates, we do observe that women are more satisfied with their jobs (Model 2). With the additional measure of job characteristics, gender differences in job satisfaction become slightly larger (Model 3), while no statistical difference is observed in the coefficients of the gender variable (Model 2 vs Model 3). Our findings confirm that men are as happy in their workplace as women on average because of better work conditions such as a higher salary and professional occupation, but women report higher job satisfaction in identical jobs.

As mentioned earlier, due to being doubly disadvantaged (through gender and disability) [44], it appears that women with disabilities are more likely to be in a poorer position in the labor market than disabled men. Nevertheless, they report higher job satisfaction, which

**Table 1. Summary statistics.**

| Variable | Males (N = 4,543) | | Females (N = 1,528) | |
|---|---|---|---|---|
| | Mean | STD | Mean | STD |
| Job satisfaction | 3.14 | 0.65 | 3.17 | 0.60 |
| Age | 49.59 | 9.64 | 50.25 | 9.80 |
| age> = 18 & age<26 | 0.02 | 0.13 | 0.02 | 0.15 |
| age> = 26 & age<36 | 0.08 | 0.27 | 0.06 | 0.24 |
| age> = 36 & age<46 | 0.21 | 0.41 | 0.19 | 0.39 |
| age> = 46 & age<56 | 0.39 | 0.49 | 0.39 | 0.49 |
| age> = 56 & age< = 65 | 0.31 | 0.46 | 0.34 | 0.47 |
| Education | | | | |
| Primary or less | 0.24 | 0.43 | 0.40 | 0.49 |
| Middle school | 0.19 | 0.40 | 0.15 | 0.36 |
| High school | 0.40 | 0.49 | 0.34 | 0.47 |
| Collage or more | 0.17 | 0.38 | 0.11 | 0.31 |
| Married or having a partner (= 1) | 0.71 | 0.45 | 0.53 | 0.50 |
| Regions | | | | |
| Seoul | 0.16 | 0.37 | 0.21 | 0.41 |
| Incheon/Gyeonggi-do | 0.27 | 0.44 | 0.20 | 0.40 |
| Kangwon-do | 0.05 | 0.23 | 0.05 | 0.22 |
| Kyeongsang-do | 0.27 | 0.45 | 0.30 | 0.46 |
| Jeolla-do | 0.13 | 0.34 | 0.11 | 0.32 |
| Chungcheong-do | 0.11 | 0.31 | 0.13 | 0.33 |
| Disability type | | | | |
| Physical external | 0.65 | 0.48 | 0.55 | 0.50 |
| Sensory | 0.26 | 0.44 | 0.38 | 0.49 |
| Mental | 0.05 | 0.21 | 0.03 | 0.18 |
| Physical internal | 0.04 | 0.20 | 0.04 | 0.20 |
| Severe disability (= 1) | 0.26 | 0.44 | 0.24 | 0.43 |
| Disability onset age | 24.95 | 16.79 | 21.93 | 19.03 |
| Daily working hours | 9.29 | 3.38 | 7.78 | 2.09 |
| Monthly wage | 152.10 | 96.67 | 89.43 | 61.85 |
| Regular worker (= 1) | 0.42 | 0.49 | 0.27 | 0.44 |
| Establishment size | | | | |
| 1–24 | 0.60 | 0.49 | 0.63 | 0.48 |
| 25–299 | 0.27 | 0.44 | 0.28 | 0.45 |
| 300–999 | 0.05 | 0.22 | 0.04 | 0.19 |
| more than 1,000 | 0.08 | 0.27 | 0.05 | 0.21 |
| Occupations | | | | |
| Managers | 0.02 | 0.14 | 0.001 | 0.03 |
| Professionals | 0.05 | 0.22 | 0.04 | 0.19 |
| Associate Professionals | 0.10 | 0.31 | 0.10 | 0.30 |
| Clerical | 0.04 | 0.21 | 0.17 | 0.38 |
| Service/sales | 0.03 | 0.18 | 0.06 | 0.23 |
| Farmers/fishers | 0.01 | 0.09 | 0.02 | 0.13 |
| Craft/trade | 0.20 | 0.40 | 0.05 | 0.21 |
| Manufacturer | 0.12 | 0.33 | 0.02 | 0.15 |
| Low skilled/simple work | 0.42 | 0.49 | 0.55 | 0.50 |
| Industry | | | | |

*(Continued)*

**Table 1.** (Continued)

| Variable | Males (N = 4,543) | | Females (N = 1,528) | |
|---|---|---|---|---|
| | Mean | STD | Mean | STD |
| Agriculture | 0.02 | 0.14 | 0.04 | 0.20 |
| Mining/utility | 0.02 | 0.15 | 0.00 | 0.07 |
| Manufacturing | 0.24 | 0.43 | 0.24 | 0.43 |
| Construction | 0.18 | 0.38 | 0.01 | 0.09 |
| Wholesale/trade/accommodation | 0.16 | 0.37 | 0.18 | 0.38 |
| Information/communication | 0.02 | 0.15 | 0.01 | 0.10 |
| Finance/insurance | 0.02 | 0.14 | 0.01 | 0.12 |
| Real estate | 0.03 | 0.16 | 0.02 | 0.13 |
| Professional/support service | 0.08 | 0.27 | 0.08 | 0.27 |
| Public/education/health | 0.15 | 0.36 | 0.28 | 0.45 |
| Other service | 0.08 | 0.27 | 0.13 | 0.34 |

conforms to gender difference in job satisfaction among nondisabled workers [1, 9, 10]. However, our findings are inconsistent with those of Pagán and Malo [12], who reported no difference in job satisfaction between genders among workers with disabilities in Spain.

### 4.3. Work values

The last column in Table 3 shows the estimation results where a set of variables of work values is added. With the reference group 'pay' excluded, the rest of the indicator variables are included to allow for the role of work values in explaining job satisfaction. Workers who place weight on the intrinsic aspects of a job—aptitude/desire, job status/security, or work reputation/size—report significantly higher job satisfaction than those who value pay. We also observe, with the variables of work values added, that the gender differentials in job satisfaction decreases statistically insignificant from 0.225 (Model 3) to 0.222 (Model 4). The gender gap in job satisfaction remains considerable, however, implying that there could be other factors unobserved by the researchers but associated with job satisfaction.

Additional tests were conducted in relation to the models, and the results are shown in Table 3. By testing the null hypothesis of $H_0 : \sigma_v^2 = 0$ through the likelihood ratio test, we confirmed that a random-effects model is more desirable than a pooled one. The overall

**Table 2. The most important aspects of work.**

| Work values | Percentage of the most important factor | | |
|---|---|---|---|
| | Men | Women | t-statistic |
| Pay | 50.76 | 43.78 | 4.11 |
| Work intensity | 6.78 | 9.49 | -3.49 |
| Work hours | 1.43 | 3.60 | -5.28 |
| Aptitude/desire | 26.85 | 31.02 | -3.14 |
| Job status/security | 10.46 | 8.25 | 2.50 |
| Disability accommodation | 1.63 | 1.11 | 1.44 |
| Relations at work | 0.31 | 0.20 | 0.72 |
| Benefits | 0.22 | 0.20 | 0.17 |
| Commuting distance/work flexibilities | 0.92 | 1.90 | -3.06 |
| Success potential | 0.42 | 0.07 | 2.08 |
| Work reputation/size | 0.22 | 0.39 | -1.14 |

**Table 3. Coefficient estimates of job satisfaction function.**

| Variables | Model 1 | Model 2 | Model 3 | Model 4 |
|---|---|---|---|---|
| Female (= 1) | 0.115 | 0.212*** | 0.225*** | 0.222*** |
| | (0.075) | (0.074) | (0.073) | (0.073) |
| Age | | -0.019 | -0.037* | -0.039* |
| | | (0.023) | (0.022) | (0.022) |
| Age squared/1000 | | 0.000 | 0.000** | 0.000** |
| | | (0.000) | (0.000) | (0.000) |
| Middle school | | 0.098 | 0.014 | 0.004 |
| | | (0.095) | (0.087) | (0.087) |
| High school | | 0.506*** | 0.167** | 0.154* |
| | | (0.087) | (0.081) | (0.081) |
| Collage or more | | 0.876*** | -0.037 | -0.062 |
| | | (0.114) | (0.116) | (0.116) |
| Married/having a partner (= 1) | | 0.139** | 0.034 | 0.033 |
| | | (0.068) | (0.063) | (0.063) |
| Sensible | | -0.206*** | -0.199*** | -0.208*** |
| | | (0.075) | (0.068) | (0.068) |
| Mental | | -0.265 | 0.313** | 0.289* |
| | | (0.168) | (0.159) | (0.159) |
| Physical internal | | -0.072 | -0.034 | -0.033 |
| | | (0.159) | (0.144) | (0.144) |
| Severe disability (= 1) | | 0.073 | 0.098 | 0.090 |
| | | (0.079) | (0.073) | (0.073) |
| Disability onset age | | -0.001 | -0.003 | -0.003 |
| | | (0.002) | (0.002) | (0.002) |
| Daily working hours | | | -0.022** | -0.021** |
| | | | (0.008) | (0.008) |
| Log income | | | 0.467*** | 0.473*** |
| | | | (0.047) | (0.047) |
| Regular worker (= 1) | | | 0.221*** | 0.221*** |
| | | | (0.066) | (0.066) |
| Year | No | Yes | Yes | Yes |
| Residential district | No | Yes | Yes | Yes |
| Company size | No | No | Yes | Yes |
| Occupation | No | No | Yes | Yes |
| Industry | No | No | Yes | Yes |
| Work values | No | No | No | Yes |
| $\kappa_1$ | -3.308*** | -3.792*** | -2.612*** | -2.580*** |
| | (0.081) | (0.569) | (0.619) | (0.619) |
| $\kappa_2$ | -1.687*** | -2.168*** | -0.979 | -0.942 |
| | (0.048) | (0.564) | (0.616) | (0.616) |
| $\kappa_3$ | 0.921*** | 0.453 | 1.666*** | 1.710*** |
| | (0.042) | (0.563) | (0.615) | (0.616) |
| $\kappa_4$ | 3.771*** | 3.320*** | 4.603*** | 4.656*** |
| | (0.103) | (0.572) | (0.626) | (0.626) |
| $\sigma_v^2$ | 1.122*** | 0.973*** | 0.710*** | 0.703*** |
| | (0.073) | (0.066) | (0.054) | (0.053) |
| Wald_chi2 | 2.325 | 189.604 | 585.192 | 609.733 |

(*Continued*)

**Table 3.** (Continued)

| Variables | Model 1 | Model 2 | Model 3 | Model 4 |
|---|---|---|---|---|
| LR test | 1,388.089 | 1,157.665 | 746.659 | 733.257 |
| Hit-miss ratio | 0.615 | 0.617 | 0.645 | 0.647 |
| Number of observations | 6,071 | | | |

Notes:

*** = p<0.01,

** = p<0.05,

* = p<0.10. Complete results available from the authors.

Source: PSED from 2009 to 2015. The values in parenthesis represent standard error.

significance of the models estimated was checked through the Wald test, confirming that unrestricted models in the three different specifications are valid. Finally, we measured the hit-miss ratio to see how well the estimated model predicts the actual dependent variable. On the basis of the correctly predicted percentages, it seems appropriate to apply the random-effects model, and the models estimated in this study are considered to be generally significant.

However, the magnitude of an estimated coefficient of an ordered probit model is not interpretable directly because the marginal effect is a function of more than the estimated coefficients [47]. Thus, the estimated coefficients have been supplemented by reporting the marginal effects of the gender dummy in Table 4. In Model 2, consistent with the earlier results, the marginal effects show that, on average, females are 2.4 percentage points less likely than males to report their job satisfaction as "dissatisfied," and about 4.6 percentage points more likely to say they are "satisfied." Despite the different model specifications, with the exception of Model 1, which features unconditional gender difference, the overall observations remain constant, with slight differences.

## 4.4. Expectations

To explain the significant gender difference in job satisfaction, we propose the presence of gender difference in job expectations: Women expect less from their job than men do, given the identical job, leading to higher job satisfaction among women. Satisfaction may be determined partly by relative factors rather than absolute ones. The process of comparison may take place

**Table 4. Marginal effect of gender variable.**

| | Model1 | Model2 | Model3 | Model4 |
|---|---|---|---|---|
| Strongly dissatisfied (= 1) | -0.002 | -0.004 | -0.004 | -0.004 |
| | (0.002) | (0.002) | (0.001) | (0.001) |
| Dissatisfied (= 2) | -0.013 | -0.024 | -0.026 | -0.026 |
| | (0.009) | (0.009) | (0.008) | (0.008) |
| Neutral (= 3) | -0.010 | -0.020 | -0.020 | -0.020 |
| | (0.007) | (0.007) | (0.007) | (0.007) |
| Satisfied (= 4) | 0.025 | 0.046 | 0.048 | 0.047 |
| | (0.016) | (0.016) | (0.016) | (0.016) |
| Strongly satisfied (= 5) | 0.001 | 0.002 | 0.002 | 0.002 |
| | (0.001) | (0.001) | (0.001) | (0.001) |

Notes: The values in parenthesis represent standard error.

across a range of personal and job attributes. Through indirect approaches, we examined a set of variables that may be correlated with an individual's job expectation [2].

The first test is to examine the association between gender difference in job satisfaction and age and education. We anticipated that gender differentials in expectations will be lower for younger workers, who are usually more ambitious, and for better-educated workers, who evaluate themselves as more valuable in the labor market [3, 8]. Job expectations may also be changed by subsequent labor-market experience. In particular, exposure to high-quality jobs might raise women's lower expectations upward. Thus, it is supposed that the gender gap in job satisfaction will be lower across professional occupations.

To investigate the role of gender, we re-ran the regression analysis with interaction between gender and age, and we conducted separate regression analyses, as shown in Table 5 (Model 1). We observed that the difference in job satisfaction between gender is lower for workers aged 18 to 35, but we are not able to reject the null hypothesis that there is a gender difference in job satisfaction between the junior and the senior group. A very similar result is observed among college graduates (Model 2). However, we find a hypothesis-consistent result related to educational attainment (Model 3): While women report significantly higher job satisfaction, holding other factors constant, the difference in satisfaction considerably diminishes among the highly educated group. Our findings remain constant in the comprehensive model (Model 4): It appears that women are happier at work but that the difference in satisfaction between genders is lower among college graduates. While the sum of the three different coefficients of interaction terms is large enough in absolute terms to offset the coefficient estimate of the gender variable, the interaction between gender and education is the only one with statistical significance.

Assuming that the workers mentioned above—younger, highly educated, and with professional occupations—are likely to have similar expectations for males and females, we conclude that the lower expectations of female workers with disabilities are likely to play a limited role in explaining their higher job satisfaction relative to their male counterparts. While the results of testing the role of expectation provided some weak evidence for gender differences in job satisfaction among workers with disabilities, our data do not fully support the hypothesis applicable to the gender differences in job satisfaction for nondisabled workers [1, 2, 4, 23].

## 5. Conclusion

Using the PSED, which restricts the sample to the population with disabilities, this article analyzed the association of job satisfaction with gender among disabled workers. We considered the role of both objective and subjective variables—personal and job characteristics, work values, and expectations—as determinants of the gender gap in job satisfaction among workers with disabilities. Higher levels of job satisfaction were observed for disabled women relative to men with disabilities, holding personal and job-related characteristics fixed. Having added work values into the specification of the job satisfaction function, the gender differences in job satisfaction marginally decreased, but the change was not statistically significant.

We ascribed a part of the gender difference in job satisfaction to the gender gap in expectations from jobs. The hypothesis that there exists little difference in job satisfaction within a more homogeneous group was *partially* supported by empirical evidence that the gender difference in job satisfaction diminished when restricting the samples to subsamples, such as younger workers, highly educated workers, or workers in high-quality jobs. The results of this study are largely consistent with those of previous studies on the job satisfaction of nondisabled people. However, it appeared that the hypothesis related to expectations that was applied

**Table 5. Testing the role of expectation.**

| Variables | Model 1 | Model 2 | Model 3 | Model 4 |
|---|---|---|---|---|
| Female (= 1) | 0.268*** | 0.404*** | 0.264*** | 0.497*** |
| | (0.075) | (0.100) | (0.076) | (0.116) |
| Age 18–35 (= 1) | 0.018 | -0.018 | -0.009 | 0.040 |
| | (0.103) | (0.094) | (0.094) | (0.103) |
| Age 18–35 * Sex | -0.142 | | | -0.251 |
| | (0.205) | | | (0.214) |
| Professional, managerial (= 1) | 0.365*** | 0.361*** | 0.378*** | 0.404*** |
| | (0.076) | (0.076) | (0.084) | (0.084) |
| Professional, managerial * Sex | | -0.074 | | -0.170 |
| | | (0.167) | | (0.175) |
| College or more (= 1) | -0.101 | -0.023 | -0.099 | -0.007 |
| | (0.065) | (0.073) | (0.065) | (0.074) |
| College or more * Sex | | | -0.288** | -0.370*** |
| | | | (0.133) | (0.142) |
| Married/having a partner (= 1) | 0.045 | 0.039 | 0.046 | 0.036 |
| | (0.063) | (0.063) | (0.063) | (0.063) |
| Sensible | -0.187*** | -0.194*** | -0.187*** | -0.193*** |
| | (0.068) | (0.068) | (0.068) | (0.068) |
| Mental | 0.250 | 0.248 | 0.249 | 0.244 |
| | (0.156) | (0.156) | (0.156) | (0.156) |
| Physical internal | -0.061 | -0.065 | -0.057 | -0.068 |
| | (0.145) | (0.145) | (0.145) | (0.144) |
| Severe disability (= 1) | 0.097 | 0.087 | 0.094 | 0.093 |
| | (0.073) | (0.073) | (0.073) | (0.073) |
| Disability onset age | -0.001 | -0.001 | -0.001 | -0.001 |
| | (0.002) | (0.002) | (0.002) | (0.002) |
| Daily working hours | -0.019** | -0.020** | -0.019** | -0.019** |
| | (0.008) | (0.008) | (0.008) | (0.008) |
| Log income | 0.448*** | 0.448*** | 0.449*** | 0.449*** |
| | (0.046) | (0.046) | (0.046) | (0.046) |
| Regular worker (= 1) | 0.211*** | 0.216*** | 0.210*** | 0.216*** |
| | (0.065) | (0.065) | (0.065) | (0.065) |
| Year | Yes | Yes | Yes | Yes |
| Residential district | Yes | Yes | Yes | Yes |
| Company size | Yes | Yes | Yes | Yes |
| Industry | Yes | Yes | Yes | Yes |
| Work values | Yes | Yes | Yes | Yes |
| $\kappa_1$ | -1.394*** | -1.400*** | -1.399*** | -1.372*** |
| | (0.290) | (0.289) | (0.290) | (0.289) |
| $\kappa_2$ | 0.243 | 0.234 | 0.237 | 0.261 |
| | (0.285) | (0.284) | (0.285) | (0.285) |
| $\kappa_3$ | 2.893*** | 2.884*** | 2.886*** | 2.911*** |
| | (0.288) | (0.288) | (0.288) | (0.288) |
| $\kappa_4$ | 5.838*** | 5.830*** | 5.832*** | 5.858*** |
| | (0.309) | (0.309) | (0.309) | (0.309) |
| $\sigma_v^2$ | 0.721*** | 0.717*** | 0.720*** | 0.714*** |
| | (0.054) | (0.054) | (0.054) | (0.054) |

(*Continued*)

**Table 5.** (Continued)

| Variables | Model 1 | Model 2 | Model 3 | Model 4 |
|---|---|---|---|---|
| Number of observations | 6,071 | | | |

Notes:

*** = p<0.01,

** = p<0.05,

* = p<0.10. Complete results available from the authors. Source: PSED from 2009 to 2015. The values in parenthesis represent standard error.

when explaining the gender–job satisfaction gap among nondisabled workers could not be fully applied to workers with disabilities.

In Korea, the employment rate and mean wage of disabled women are reported to be around half of those of disabled men [48]. This allows us to reasonably infer that the poor labor-market position of women with disabilities plays a major role in forming a lower expectation. Hence, the government needs to consider social policies to improve the adverse conditions faced by disabled women in the labor market (e.g., a mandatory employment quota system and supplementation of wages).

We acknowledge some limitations in our research. We were not able to reject some of the null hypotheses, but failing to reject the null hypothesis does not mean that there are no gender differences in job satisfaction; they are unlikely to have enough power to reject the null hypothesis because of the small sample size. This paper carried out an indirect test of the relative utility hypothesis that expectations explain the gender difference in job satisfaction among a population with disabilities. No definitive test of the expectation variable can be applied because reliable information on expectations is rarely collected. Future research may contribute to the literature by developing an alternative measure of expectation, which would enable us to investigate the role of expectation in determining job satisfaction more directly. One possibility is that a future survey could ask workers about their level of expectation from their jobs in terms of pay, job security, work hours, or work environment, for example.

## Author Contributions

**Methodology:** Seunghee Yu.

**Writing – original draft:** Chung Choe.

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
