## [Decision Letter · Decision Letter 0]

13 Nov 2020

PONE-D-20-18796

Gender Differences in Job Satisfaction among Disabled Workers

PLOS ONE

Dear Dr. Choe,

Thank you for submitting your manuscript to PLOS ONE. After careful consideration, we feel that it has merit but does not fully meet PLOS ONE’s publication criteria as it currently stands. Therefore, we invite you to submit a revised version of the manuscript that addresses the points raised during the review process.

We look forward to receiving your revised manuscript.

Kind regards,

M Niaz Asadullah

Academic Editor

PLOS ONE

Journal Requirements:

Reviewers' comments:

Reviewer's Responses to Questions

**Comments to the Author**

1. Is the manuscript technically sound, and do the data support the conclusions?

Reviewer #1: Yes

Reviewer #2: Partly

2. Has the statistical analysis been performed appropriately and rigorously? 

Reviewer #1: No

Reviewer #2: No

3. Have the authors made all data underlying the findings in their manuscript fully available?

Reviewer #1: Yes

Reviewer #2: No

4. Is the manuscript presented in an intelligible fashion and written in standard English?

Reviewer #1: Yes

Reviewer #2: Yes

5. Review Comments to the Author

Reviewer #1: Using a panel data from PSED, this paper examines the effects of gender on job satisfaction among disable workers in Korea. This paper addresses the gender differences in job satisfaction of disable workers which is an important and current issue in the literature of job/life satisfaction. It is a well-structured paper, with a steady focus – the gender differences are explored using different model specifications, including the control variables of work values and expectations (proxy). Most of my doubts and questions tend to be addressed quickly. This paper has a potential benefit to the readers of PLOS ONE. Nevertheless, this paper is not publishable in its present form due to the some major concerns (pls refer to the uploaded reviewer report for details).

Reviewer #2: The paper examines gender differences in job satisfaction among disabled workers using random-effects ordered-probit models. The authors follow Clark (1997) to some extent. The main contribution of the paper is an analysis of this gender differences among disabled workers.

1. I wonder why the authors exclude “observations with missing information for any relevant variables”. Is it to have a balanced panel dataset? The problem is they drop most of the observations. For the year 2015, for example, they keep only 19% of observations in the dataset. I think the authors should use larger sample size if possible. Having unbalanced panel data or fewer independent variables is fine to have a more representative sample. It will also increase the sample size so that the authors have larger power to reject the null hypotheses.

2. The authors estimate separate regressions by age, education, and occupation to explore whether expectations explain gender differences in job satisfaction whose results are presented in Table 5. I think it is difficult to draw conclusions from Table 5 because (1) whether the estimates differ is not formally tested; (2) the authors may have weak power to reject the null hypothesis in some of the specifications (the sample size is small); (3) the authors cannot draw conclusions by comparing the statistical significance (in fact, the difference between statistically significant and statistically insignificant estimates may be statistically insignificant). I think the authors should estimate regressions with interactions between gender and each of the above three variables; I think that is the more proper approach.

3. The authors seem to only cite Clark (1997) to justify their approach of testing whether expectations explain gender differences in job satisfaction. Consider elaborating this approach further so that readers can see why this approach may help answering the research question.

4. I think the authors should more carefully draw conclusions from their results. In the abstract, for example, they conclude “[their] findings reveal that different work values between women and men do not account for the still significant higher job satisfaction of women, although they contribute to a partial decrease of the gender gap in job satisfaction”. I do not see evidence that different work values “contribute to a partial decrease of the gender gap” in Table 4. (The estimate “decreases” from 0.225 to 0.222 but probably they do not differ statistically.) I am also unsure what they mean when they say job satisfaction of women is significantly higher; do they mean significantly higher statistically? They then conclude that the hypothesis on the role of expectations is “supported by the empirical analyses that gender differentials disappear for the young, …”. But failing to reject the null hypothesis does not mean that there are no gender differences in job satisfaction; maybe they do not have enough power to reject the null hypothesis because the sample size is smaller. Moreover, as I write above, the difference between gender differences in job satisfaction in the “young” and “older” specifications may be statistically insignificant. (I find other possible misinterpretations of their results: I do not see evidence that adding measures of job characteristics increases gender differences in job satisfaction (Table 2); I do not see evidence that “women are more likely to put more weight on work intensity, work hours, aptitude, …” (Table 3); job satisfaction may not necessarily decrease just because the estimate in Model 4 is smaller; see also the conclusions.)

5. Consider using fixed-effects in addition to random-effects ordered probit model to see whether the results are robust.

6. Consider presenting the standard errors in tables of results, not only the asterisks.

6. PLOS authors have the option to publish the peer review history of their article (what does this mean?). If published, this will include your full peer review and any attached files.

Reviewer #1: **Yes: **Hock-Eam Lim

Reviewer #2: No

---

## [Decision Letter · Decision Letter 1]

6 May 2021

PONE-D-20-18796R1

Gender Differences in Job Satisfaction among Disabled Workers

PLOS ONE

Dear Authors,

Thank you for submitting your manuscript to PLOS ONE. After careful consideration, we feel that it has merit but does not fully meet PLOS ONE’s publication criteria as it currently stands. Therefore, we invite you to submit a revised version of the manuscript that addresses the points raised during the review process.

We look forward to receiving your revised manuscript.

Kind regards,

Hafiz T.A. Khan, Ph.D, CStat

Academic Editor

PLOS ONE

Journal Requirements:

Reviewers' comments:

Reviewer's Responses to Questions

**Comments to the Author**

1. If the authors have adequately addressed your comments raised in a previous round of review and you feel that this manuscript is now acceptable for publication, you may indicate that here to bypass the “Comments to the Author” section, enter your conflict of interest statement in the “Confidential to Editor” section, and submit your "Accept" recommendation.

Reviewer #1: (No Response)

Reviewer #3: All comments have been addressed

2. Is the manuscript technically sound, and do the data support the conclusions?

Reviewer #1: Yes

Reviewer #3: Yes

3. Has the statistical analysis been performed appropriately and rigorously? 

Reviewer #1: Yes

Reviewer #3: Yes

4. Have the authors made all data underlying the findings in their manuscript fully available?

Reviewer #1: No

Reviewer #3: Yes

5. Is the manuscript presented in an intelligible fashion and written in standard English?

Reviewer #1: Yes

Reviewer #3: Yes

6. Review Comments to the Author

Reviewer #1: The revised paper has improved substantially and most of the concerns are addressed satisfactorily. This paper indeed has a potential benefit to the readers of PLOS ONE. There are only a few minor concerns which the authors might wish to address with:

The variable, 〖JS〗_it^ , need to be labelled as well (line 245). For instance, “…probability that 〖JS〗_it^ (i.e., the reported job satisfaction) is equivalent …”

The specification of cut points seems inaccurate. “…κ is a set of cut points κ_1,κ_2,…,κ_4 ; …” (line 250). This specification does not match with the Equation (3) in terms of j. According to Long (1997), the specification should be “…κ is a set of cut points κ_0,κ_2,…,κ_5 (where κ_0=-∞ & κ_5=+∞) ; …”. This will match the Equation (3).

It is stated that “…Our findings confirm that men are as happy in their workplace as women on average …” (line 324-325). From Table 3, it shows that women are happier than men. Please check.

The use of the words, “…statistically insignificantly…” (line 340) seems weird. Please rewrite this sentence.

The LR test statistics in Table 3 (Model 1) seem to incorrectly report. The extremely high values of LR test statistics (1388.089) appears to contradict with the low Wald chi-squared test statistics (2.325). Both LR test and Wald test are overall fit tests. They normally would not have huge differences in their test statistics. Please check.

Title of Table 3, 4 and 5 include the word, “standard error” in parenthesis. How about delete it and include a footnote under the tables, such as “The values in parenthesis represent standard error”?

Please refer to the attached reviewer report for a better presentation of the characters and symbols.

Reviewer #3: The previous comments by Reviewer 1 and Reviewer 2 are robustly addressed by the authors. This significantly made the results and findings more clearer and unambiguous. However, the concerns of the Reviewer 2 (Comments 2.1 and 2.4) on smaller size of the sample is an important issue which needs to be acknowledged explicitly as a research limitation in the conclusions section.

7. PLOS authors have the option to publish the peer review history of their article (what does this mean?). If published, this will include your full peer review and any attached files.

Reviewer #1: **Yes: **Lim Hock-Eam

Reviewer #3: No

---

## [Author Response · Author response to Decision Letter 1]

9 May 2021

Reviewer #1

The revised paper has improved substantially and most of the concerns are addressed satisfactorily. This paper indeed has a potential benefit to the readers of PLOS ONE. There are only a few minor concerns which the authors might wish to address with:

The variable, 〖JS〗_it^ , need to be labelled as well (line 245). For instance, “…probability that 〖JS〗_it^ (i.e., the reported job satisfaction) is equivalent …”

The specification of cut points seems inaccurate. “…κ is a set of cut points κ_1,κ_2,…,κ_4 ; …” (line 250). This specification does not match with the Equation (3) in terms of j. According to Long (1997), the specification should be “…κ is a set of cut points κ_0,κ_2,…,κ_5 (where κ_0=-∞ & κ_5=+∞) ; …”. This will match the Equation (3).

It is stated that “…Our findings confirm that men are as happy in their workplace as women on average …” (line 324-325). From Table 3, it shows that women are happier than men. Please check.

The use of the words, “…statistically insignificantly…” (line 340) seems weird. Please rewrite this sentence.

The LR test statistics in Table 3 (Model 1) seem to incorrectly report. The extremely high values of LR test statistics (1388.089) appears to contradict with the low Wald chi-squared test statistics (2.325). Both LR test and Wald test are overall fit tests. They normally would not have huge differences in their test statistics. Please check.

Title of Table 3, 4 and 5 include the word, “standard error” in parenthesis. How about delete it and include a footnote under the tables, such as “The values in parenthesis represent standard error”?

Response: We addressed all of the comments raised by the reviewer above. We appreciate the referee for reviewing our paper very carefully. One thing we would like to note is that, following the suggestion, we did double-check the low Wald chi-squared test statistics by re-running our statistical package, but we are sure that it was the correct result. Lastly, in terms of the interpretation of Table 3, while Model 1 does not show gender difference, we observe the gender difference in Model 2 and 3. This is why we state “men are as happy in their workplace as women on average because of better work conditions such as a higher salary and professional occupation, but women report higher job satisfaction in identical jobs”. 

Reviewer #3

The previous comments by Reviewer 1 and Reviewer 2 are robustly addressed by the authors. This significantly made the results and findings more clearer and unambiguous. However, the concerns of the Reviewer 2 (Comments 2.1 and 2.4) on smaller size of the sample is an important issue which needs to be acknowledged explicitly as a research limitation in the conclusions section.

Response: Thank you for the comment. We added the research limitation in the conclusion section as below. 

“We acknowledge some limitations in our research. We were not able to reject some of the null hypotheses, but failing to reject the null hypothesis does not mean that there are no gender differences in job satisfaction; they are unlikely to have enough power to reject the null hypothesis because of the small sample size.”

---

## [Editor Report · Decision Letter 2]

14 May 2021

Gender Differences in Job Satisfaction among Disabled Workers

PONE-D-20-18796R2

Dear Authors,

We’re pleased to inform you that your manuscript has been judged scientifically suitable for publication and will be formally accepted for publication once it meets all outstanding technical requirements.

Kind regards,

Professor Hafiz T.A. Khan, Ph.D, CStat

Academic Editor

PLOS ONE
---

## [Editor Report · Acceptance letter]

27 May 2021

PONE-D-20-18796R2 

Gender differences in job satisfaction among disabled workers 

Dear Dr. Choe:

I'm pleased to inform you that your manuscript has been deemed suitable for publication in PLOS ONE. Congratulations! Your manuscript is now with our production department. 

Kind regards, 

on behalf of

Professor Hafiz T.A. Khan 

Academic Editor

PLOS ONE